# One-Step Synthesis of a Non-Precious-Metal Tris (Fe/N/F)-Doped Carbon Catalyst for Oxygen Reduction Reactions

**DOI:** 10.3390/molecules28052392

**Published:** 2023-03-05

**Authors:** Huitian Yang, Hao Wu, Lei Yao, Siyan Liu, Lu Yang, Jieling Lu, Hongliang Peng, Xiangcheng Lin, Ping Cai, Huanzhi Zhang, Fen Xu, Kexiang Zhang, Lixian Sun

**Affiliations:** Guangxi Key Laboratory of Information Material, Guangxi Collaborative Innovation Center of Structure and Property for New Energy and Materials, School of Material Science and Engineering, Guilin University of Electronic Technology, Guilin 541004, China

**Keywords:** doped carbon materials, non-precious-metal catalysts, oxygen reduction reaction, electrocatalysts, fuel cells

## Abstract

Advancements in inexpensive, efficient, and durable oxygen reduction catalysts is important for maintaining the sustainable development of fuel cells. Although doping carbon materials with transition metals or heteroatomic doping is inexpensive and enhances the electrocatalytic performance of the catalyst, because the charge distribution on its surface is adjusted, the development of a simple method for the synthesis of doped carbon materials remains challenging. Here, a non-precious-metal tris (Fe/N/F)-doped particulate porous carbon material (2_1_P_2_-Fe_1_-850) was synthesized by employing a one-step process, using 2-methylimidazole, polytetrafluoroethylene, and FeCl_3_ as raw materials. The synthesized catalyst exhibited a good oxygen reduction reaction performance with a half-wave potential of 0.85 V in an alkaline medium (compared with 0.84 V of commercial Pt/C). Moreover, it had better stability and methanol resistance than Pt/C. This was mainly attributed to the effect of the tris (Fe/N/F)-doped carbon material on the morphology and chemical composition of the catalyst, thereby enhancing the catalyst’s oxygen reduction reaction properties. This work provides a versatile method for the gentle and rapid synthesis of highly electronegative heteroatoms and transition metal co-doped carbon materials.

## 1. Introduction

With the increasing environmental awareness, the development of new energy storage and conversion technologies that can effectively reduce climate pollution is currently gaining attention [1,2,3,4,5]. Fuel cells are environmentally-friendly energy devices that can directly convert the energy released from a chemical reaction into electrical energy [6]. These have the advantages of high energy conversion efficiency, ease of use, and being environmentally friendly. Hence, these are considered to be one of the most promising clean energy conversion technologies [7,8,9]. A crucial reaction in fuel cells and metal–air cells is the oxygen reduction reaction (ORR) [10]. Currently, precious metals are used as catalysts for the ORR, owing to the low stability and durability of non-precious-metal catalysts [11,12]. Pt is considered the most suitable catalyst for the ORR [13,14]. However, platinum is expensive and scarce. In addition, Pt-based catalysts can be deactivated or poisoned during electrocatalytic processes [15,16,17,18,19]. Consequently, numerous studies on platinum-free ORR catalysts have stalled. Thus, high-performance, inexpensive, multi-element-doping platinum-free catalysts or novel low-platinum catalysts should be explored [20,21].

Studies have shown that the catalyst performance can be adjusted with (1) specific heteroatoms having an electronegativity higher than that of carbon but with a similar atomic radius, which facilitates the doping into the lattice of carbon materials [22], and (2) transition metals with empty 3d orbitals [23]. The heteroatom and transition metal co-doped carbon materials can effectively adjust the charge distribution on the surface of carbon materials, thus achieving the adjustment in the catalyst performance [24]. Among the elements used in the heteroatom-doped carbon catalysts, nitrogen (N), oxygen (O), and fluorine (F) are considered to be excellent doping elements in carbon materials because of their high electronegativity and an atomic radius similar to that of carbon [25]. Catalysts with nitrogen-doped carbon-based structures (NC structures) have been extensively studied, mainly because they outperform commercial Pt/C catalysts in alkaline environments [26]. Oxygen atoms are usually present in heat-treated carbon materials, whereas N and F must be doped into carbon materials using special doping methods. Inorganic fluorine salts (such as NaF and NH_4_F) are commonly used for the F doping of carbon materials, although they pose severe environmental concerns [27]. Fluoride doping is gentle and efficient for doping organic matters, such as polytetrafluoroethylene (PTFE) and polyvinylidene fluoride [28].

Iron (Fe) and cobalt (Co) atoms with 3d vacant orbitals are believed to be excellent transition metal promoters for NC structure formation [29,30,31,32,33]. In particular, iron has a high annual yield and is inexpensive [34]; therefore, Fe-N_x_/C structured catalysts are of considerable importance for the ORR.

However, obtaining N, F, and Fe co-doped carbon catalysts with excellent electrochemical properties using a simple synthetic method remains challenging.

In this study, we used a one-step method with simple raw materials to synthesize tris (Fe/N/F)-doped particulate porous carbon material (2_1_P_2_-Fe_1_-850). The prepared catalyst exhibited an ORR catalytic performance comparable to that of commercial Pt/C catalysts and better stability and methanol resistance, thus demonstrating its potential applications in ORRs.

## 2. Results and Discussion

As shown in Figure 1, 2P-Fe-X was synthesized using ferric chloride as the iron source, 2-methylimidazole as the nitrogen and carbon source, and PTFE as the fluorine and carbon source. The raw materials were ground for 0.5 h until they were uniformly ground. A high-temperature solid-phase pyrolysis was then carried out for 1 h (under a N_2_ atmosphere) at 850 °C to obtain a carbon nanomaterial with an F, N, and Fe triple doping. The possible mechanism of synthesis is that, at high temperatures, 2-methylimidazole and PTFE are pyrolyzed to some small molecules, which are then recombined in the presence of iron as a catalyst to form new doped carbon materials. A similar synthesis was performed without the addition of FeCl_3_, with different ratios of 2-methylimidazole and PTFE, and at different temperatures for the F and N co-doped carbon materials.

Several studies have reported that the best ORR performance can be obtained at a pyrolysis temperature between 800 and 1000 °C [35,36,37,38]. Here, the pyrolysis temperature was fixed at 900 °C, and the effect of the precursor ratio (2-methylimidazole to PTFE) on the ORR performance was investigated. When this ratio was 1:2, the half-wave potential (E_1/2_) of 2_1_P_2_ was 0.80 V (Figure 1a). E_1/2_ decreased with a further increase in the PTFE content. When the ratio of 2-methylimidazole to PTFE was 1:1–1:3, the E_1/2_ of the prepared sample was 0.75–0.80 V. This analysis confirmed that the ratio of the reactants is one of the key factors affecting the activity of the catalyst ORR and that the optimal ratio of 2-methylimidazole to PTFE is 1:2. Note that the limit diffusion currents of catalysts 2_1_P_1_ and 2_1_P_1.5_ are larger than those of other catalysts, which may be related to the structure of the samples [39].

After the optimal ratio of the precursors, 2-methylimidazole and PTFE, was determined, we further studied the effect of different temperatures on the electrocatalytic performance. Different heat-treatment processes affect the degree of graphitization, morphology, and active site composition of the carbon material [15]. The effects of the pyrolysis temperature on the catalysts are shown in Figure 1b. At a 2-methylimidazole-to-PTFE ratio of 1:2 and at a heat-treatment temperature of 700–950 °C, the value of E_1/2_ was 0.69–0.83 V, corresponding to an E_1/2_ variation of approximately 140 mV. These results indicated that the heat treatment temperature is a very important factor affecting the activity of the ORR catalyst. 2_1_P_2_-850 showed the highest ORR activity at a pyrolysis temperature of 850 °C, and its E_1/2_ was approximately 0.83 V. This is different from most studies, which have reported that optimal ORR activity of catalysts is at 900 °C. This indicates that the optimal heat-treatment temperature for different precursor mixtures is not the same because it is related to the temperature conditions between the precursors during the solid-phase synthesis of the catalyst at high temperatures.

The metal-free ORR catalyst showed good stability and was not affected by the Fenton effect [40]. Transition-metal-based ORR catalysts typically produce H_2_O_2_. In the presence of an acidic electrolyte, the dissolved transition metal ions (especially Fe^2+^) and H_2_O_2_ have strong oxidation properties (Fenton reaction), which can quickly destroy the proton exchange membrane, resulting in very poor stability of the fuel cell. However, under basic conditions, the Fenton reaction does not occur. Therefore, further doping with transition metals is an effective method for the preparation of highly efficient ORR catalysts that can be used under alkaline conditions [41]. The different metal contents in the catalyst had a significant effect on the ORR activity. The results (Figure 1c) revealed that 2_1_P_2_-Fe_0.25_-850, 2_1_P_2_-Fe_0. 5_-850, 2_1_P_2_-Fe_1_-850, 2_1_P_2_-Fe_2_-850, and 2_1_P_2_-Fe_3_-850 exhibited distinctive oxygen reduction peaks. In particular, the 2_1_P_2_-Fe_1_-850 catalyst had the largest ORR peak for current potential, indicating that its structure exposed more ORR active sites and facilitated the diffusion of oxygen.

When different volumes of FeCl_3_ (0.07 M) in the range of 0.25–3.0 mL were added to the precursor, E_1/2_ of the obtained catalyst varied between 0.81 and 0.85 V under the optimal precursor ratio and heat treatment temperature. We observed that the ORR activity of the catalysts was sensitive to the iron content; a low iron content leads to an insufficient number of active sites. If the iron content is extremely high, the iron that does not form the active site blocks the reaction channels of the catalyst or forms other catalytic sites with low ORR activity. The E_1/2_ of the catalyst 2_1_P_2_-Fe_1_-850 was approximately 0.85 V when 1.0 mL of an FeCl_3_ solution (0.07 M) was added to the precursor mixture; this is higher than the E_1/2_ of Pt/C catalysts (0.84 V).

Figure 1e,g shows the current density plots of 2_1_P_2_-850 and 2_1_P_2_-Fe_1_-850, respectively, and the C_dl_ values of the samples can be obtained based on the current density; the C_dl_ values of samples 2_1_P_2_-850 and 2_1_P_2_-Fe_1_-850 were 12.3 and 13.6 mF cm^−2^, respectively (Figure 1f,h). The electrochemical surface area (ECSA) values can be obtained according to the equation:ECSA = C_dl_/Cs (1)

The ECSA values of 2_1_P_2_-850 and 2_1_P_2_-Fe_1_-850 were 10.3 and 7.8 cm^2^, respectively.

According to the previous study, at 0.8V, the current densities of samples 2_1_P_2_-850 and 2_1_P_2_-Fe_1_-850 are 3.3 mA cm^−2^ and 4.1 mA cm^−2^, respectively. The latter is 1.2 times the former. In order to compare the intrinsic activity of 2_1_P_2_-850 and 2_1_P_2_-Fe_1_-850, we normalized the electrical density of the sample to ECSA at a potential of 0.8V. After normalization, the current density of 2_1_P_2_-850 and 2_1_P_2_-Fe_1_-850 is 0.32 mA cm^−2^ and 0.53 mA cm^−2^, respectively, and the latter is 1.65 times of the former, thus indicating that 2_1_P_2_-Fe_1_-850 possesses more active sites than 2_1_P_2_-850.

In summary, three factors affect the activity of ORR catalysts synthesized with heteroatomic and transition metal doping: (1) the proportion of heteroatomic precursors that may affect the proportion of doping atoms and the morphology of the synthesized catalyst; (2) the heat-treatment temperature that affects the ORR activity of the catalyst and plays a role in determining the structure and type of the active site of the catalyst; and (3) the transition metal co-doping and its content that can be related to the change of the catalyst active site, which can form more efficient ORR active sites with the metal.

The reaction mechanism, stability, and resistance to poisoning of ORR catalysts are the key factors reflecting the catalyst performance. To examine the kinetics of the ORR reaction of 2_1_P_2_-Fe_1_-850 catalyst, we used the polarization curves at different rotational speeds in KOH (0.1 M), which helped in elucidating the electron transfer paths. Figure 2a shows the relationship between the rotational speed and the limiting diffusion current density, where the diffusion ultimate current density (J_d_) increased with an increase in the rotational speed. This may be due to the fact that the increase in the electrode rotational speed is accompanied by a shortening of the diffusion distance of the reacting material, which facilitates the mass migration of the catalyst. Based on the Koutecky–Levich (K–L) equation, we obtained an electron transfer very close to 4.0 and with an average electron transfer number of 3.8 (Figure 2b). This indicates that 2_1_P_2_-Fe_1_-850 was dominated by a four-electron pathway.

The stability and anti-poisoning ability of ORR catalysts are crucial problems in commercial Pt/C catalysts. We performed durability and methanol tolerance tests, as shown in Figure 2c. After the 30,000 s test, the current retention of the 2_1_P_2_-Fe_1_-850 catalyst remained 92.4%, which was better than that of Pt/C (ca. 87.8%). After the addition of methanol (Figure 2d), the 2_1_P_2_-Fe_1_-850 catalyst had better methanol tolerance, and the 2_1_P_2_-Fe_1_-850 catalyst activity decreased by only 8%, whereas that of Pt/C decreased by 43%, thus showing good resistance to methanol. These results indicated that the stability and immunity to methanol poisoning of 2_1_P_2_-Fe_1_-850 were evident and comparable to those of commercial Pt/C catalysts.

Figure 3a,b and Figure 3c,d present the scanning electron microscopy (SEM) images of 2_1_P_2_-850 and 2_1_P_2_-Fe_1_-850, respectively, revealing the smooth surface of 2_1_P_2_-850. The catalyst’s smooth surface is usually not conducive to its catalytic activity, because it is not conducive to the full exposure of the catalytic site, resulting in low catalytic efficiency. The 2_1_P_2_-Fe_1_-850 sample had smaller particles on the surface, making the entire sample appear fluffier. This shows that the doping of transition metal iron can change the surface structure and morphology of heteroatom-doped carbon materials.

The transmission electron microscopy (TEM) images of 2_1_P_2_-Fe_1_-850 (Figure 3e–g revealed particle accumulation, and some small dispersed particles were observed (red ellipse markers). The scanning transmission electron microscopy (STEM) image (Figure 3h) revealed the presence of numerous pores, which were mainly formed by the accumulation of small particles. The high ORR activity of the catalyst 2_1_P_2_-Fe_1_-850, after co-doping N, F, and Fe, can be attributed to the fact that nanoparticles can fully expose the active site of the catalyst on their surface. The elemental mapping showed that we successfully introduced C, N, O, F, and Fe elements. After successful doping with Fe, F, and N, the ORR activity and stability of the catalyst may be improved by increasing the density of the active sites and fully exposing them [42].

The N_2_ adsorption–desorption curve of 2_1_P_2_-Fe_1_-850 and 2_1_P_2_-Fe_1_-850 (Figure 4a,c) reveals that it is type IV with an H3 hysteris loop, thus indicating the existence of mesopores in this material. The accumulation of irregular granular or sheet materials in this kind of material forms some meso-and macropores, and 2_1_P_2_-Fe_1_-850 has a larger number of pores than 2_1_P_2_-850. In addition, we can observe that the tail of N_2_–desorption curves of the sample 2_1_P_2_-Fe_1_-850 rapidly increases, which can be attributed to the accumulation of macroporous nanoparticles in the sample. This is consistent with the results of the TEM analysis (Figure 3).

2_1_P_2_-Fe_1_-850 has a specific surface area of 1315 m^2^ g^−1^, which is larger than that of 2_1_P_2_-850 (182 m^2^ g^−1^). This indicates that iron doping increases the specific surface area of carbon materials, thereby enhancing the materials’ exposure as electrochemical active sites. Figure 4d shows the pore size distribution curves of 2_1_P_2_-Fe_1_-850, which are consistent with the N_2_ adsorption–desorption isotherm results. We can also observe that the 2_1_P_2_-Fe_1_-850 is dominated by mesoporous pores, with the number of mesoporous pores reaching a maximum value at a pore size of 24 nm.

Figure 4e and Figure 4f show that the sizes of 2_1_P_2_-850 and 2_1_P_2_-Fe_1_-850 are concentrated at 100–600 and 200–300 nm, respectively, thus indicating that the doping of metal Fe is conducive to the reduction of the catalyst particle size. This is consistent with the SEM and X-ray diffraction (XRD) results.

Figure 5a shows the XRD spectra of 2_1_P_2_-Fe_x_-850 with different iron contents. Two broad diffraction peaks can be seen at approximately 26° and 43°, corresponding to the C (002) and C (100) crystal planes of the carbon peak, respectively. The diffraction peaks appear at around 26° and 45° when an iron solution (3 mL) is added; these are presumed to be Fe_3_C and Fe_3_O_4_ through comparison with the powder diffraction file card. The rest of the samples do not have distinctive diffraction peaks of the metallic phase, indicating that the content of the phase is extremely low and the crystal particles are considerably small and evenly distributed [43].

Figure 5b shows the XRD comparison of catalysts XC-72R, 2_1_P_1_-850, and 2_1_P_2_-Fe_1_-850, where the (002) and (100) crystal planes of the carbon peaks can be clearly observed. Compared with XC-72R (with a half-peak width of 2.6), the half-peak widths of 2_1_P_1_-850 and 2_1_P_2_-Fe_1_-850 are 3.4 and 3.5, respectively, indicating that the crystal particles of 2_1_P_2_-Fe_1_-850 are smaller and more uniformly distributed.

The Raman spectra (Figure 6) revealed that the ratio of integrated intensity in the D-band to that in the G-band (I_D_/I_G_) of 2_1_P_2_-850 was 3.7 and 3.9 in 2_1_P_2_-Fe_1_-850. This shows that the proportion of sp^3^ carbon in carbon material 2_1_P_2_-Fe_1_-850 is higher than that in carbon material 2_1_P_2_-Fe_1_-850. This indicates that sample 2_1_P_2_-Fe_1_-850 has more defects in the graphite phase structure than sample 2_1_P_2_-Fe_1_-850, resulting in a larger specific surface area and a fluffier structure. This is consistent with the previous SEM and Brunauer–Emmett–Teller (BET) results. This also indicates that iron doping is not conducive to the graphitization of carbon materials, resulting in a large number of suspended bonds in sample 2_1_P_2_-Fe_1_-850 and enhancing the catalytic activity of the ORR catalysts. In addition, our sample, 2_1_P_2_-Fe_1_-850, is difficult to graphitize at high temperatures, which may have potential applications in electrochemical energy storage materials.

Figure 7a shows the measured spectrum of the X-ray photoelectron spectroscopy (XPS) composition analysis of catalyst 2_1_P_2_-Fe_1_-850; the five elements, C, N, O, F, and Fe are shown. The content of each element was 88.41 at.%, 8.19 at.%, 2.29 at.%, 0.78 at.%, and 0.32 at.%, respectively. Figure 7b shows that the C1s peak consists of three peaks: C-C (284.8 eV), C-N (285.2 eV), and C-O (285.9 eV) [44], indicating the bonding of the doped heteroatoms with the carbon; this affected the distribution of charges on the surface of the carbon material. Figure 7c shows the N1s peak, which can be divided into four peaks at 398.4, 399.6, 401.1, and 402.3 eV. These can be classified as pyridine-N, pyrrole-N, graphite-N, and oxide-N, respectively [45], where pyridine-N, pyrrole-N, and graphite-N contribute to the “four-electron” transport [46]. The high content (92.1%) of these three Ns in 2_1_P_2_-Fe_1_-850 contributes to the ORR activity. As shown in Figure 7d, the spectrum of O1s consists of two peaks, the O-C peak at 533 eV and the O-Fe peak at 531.3 eV [29]. This indicates the formation of iron oxide in the sample, which is consistent with the previous XRD results. As shown in Figure 7e, the F1s spectrum consists of two peaks at 683.4 eV and 689.8 eV. This spectrum is noisy because the XPS signal response of F was relatively weak, and the content of F in the sample was relatively low. However, the presence of these signals indicated that F was successfully doped into the sample. This is consistent with the previous energy-dispersive X-ray spectroscopy (EDS) elemental mapping analysis. As shown in Figure 7f, the spectrum of Fe2p consists of five peaks centered at 709.9/722.4 eV and 714.0/724.1 eV, with Fe^2+^, as well as 2p3/2 and 2p1/2 states of Fe^3+^, and a satellite peak at 732 eV [47]. This indicates that the transition metal iron existed mainly in the form of Fe(II) and Fe(III) in the sample.

## 3. Materials and Methods

### 3.1. Materials

The main raw materials used were 2-methylimidazole, PTFE, and anhydrous ferric chloride. 2-methylimidazole was purchased from Shanghai Maclean Biochemical Technology Co., Ltd. (Shanghai, China). PTFE was purchased from DuPont (Wilmington, DE, USA). Anhydrous ferric chloride was purchased from Sinopharm Group Chemical Reagent Co., Ltd. (Shanghai, China).

### 3.2. Synthesis of 2_1_P_2_-Fe_1_-850 Electrocatalyst

First, anhydrous FeCl_3_ (0.58 g) was dissolved in an aqueous ethanol solution (50 mL). Next, 2-methylimidazole (3.33 g) and polytetrafluoroethylene (6.66 g) were mixed with an aqueous ethanol solution (1 mL), and the mixture was ground thoroughly in an agate mortar. The mixture was then pyrolyzed at 850 °C under a nitrogen atmosphere for 1 h. The obtained tris (Fe/N/F)-doped carbon material was named 2_1_P_2_-Fe_1_-850. Moreover, samples with different contents of FeCl_3_, different ratios of 2-methylimidazole and PTFE, and different pyrolysis temperatures were prepared by applying the same process (Table 1).

### 3.3. Physical Characteristization

XRD (D8 Advance, Bruker, Bremen, Germany) was performed using a Cu Kα light source. The composition and surface of the samples were characterized using XPS (ESCALAB 250Xi, Thermo Fisher, Waltham, MA, USA) with an Al Kα excitation source. The morphologies of the prepared samples were observed using SEM (JSM-7610Fplus, JEOL, Tokyo, Japan). TEM, high-resolution TEM, and EDS were performed using a transmission electron microscope (Talos F200X, FEI, SD, USA). Raman characterization was performed using a Raman spectrometer (Raman, Lab RAM HR Evolution, Horiba JY, Palaiseau, France) under a laser excitation of 532 nm. Nitrogen adsorption–desorption analysis was performed using a gas adsorption analyzer (BET, Autosorb iQ2, Quantachrome, Boynton Beach, FL, USA). Particle size distribution was performed using a laser particle sizer (Zetasizer Nano ZS90, Malvern, UK).

### 3.4. Electrochemical Measurements

Electrochemical tests were performed at 25 °C on a multichannel electrochemical workstation (Ivium, Eindhoven, The Netherlands), with the counter and reference electrodes being platinum wire and Hg/HgO (0.1 M KOH), respectively. A rotating disc electrode was used as the working electrode. The samples were processed as follows: the catalyst (5 mg) was added to a Nafion/ethanol solution (1 mL, 0.25 wt.%), and the mixture was sonicated for 40 min. The ink was then dispersed on a glassy carbon electrode and dried. Catalyst loading was employed as follows: the carbon material was 510 μg cm^−2^ and Pt/C was 48.9 μg Pt cm^−2^. Reversible hydrogen electrode (RHE) correction was performed as follows:E(RHE) = E(Hg/HgO) + 0.899 V (0.1 M KOH).(2)

The average number of electron transfers for the ORR was calculated using the K–L equation [47]:(3)$$\frac{1}{J}=\frac{1}{J_{k}}+\frac{1}{J_{d}}=\frac{1}{J_{k}}+B^{-1}{\sqrt{\omega }}^{-1}$$
(4)$$B=0.62\cdot n\cdot F\cdot {\left(D_{0}\right)}^{\frac{2}{3}}\cdot V^{-\frac{1}{6}}\cdot C_{0}$$
where *J* is the measured current density, *J_k_* is the kinetic current, and *J_d_* is diffusion. The constant, *B*, is the slope we determined, ω is the angular velocity on the rotating disk electrode, and *F* is the Faraday constant (96,485 mol^−1^). In 0.1 M KOH, *D*_0_ (oxygen dispersion coefficient) = 1.9 × 10^−5^ cm^2^ s^−1^, V (kinetic viscosity of the electrolyte solution) = 1.01 × 10^−2^ cm^2^ s^−1^, and *C*_0_ (volume concentration of oxygen) = 1.26 × 10^−3^ mol^−1^ [15].

## 4. Conclusions

We reported the one-step synthesis of tris (Fe/N/F)-doped carbon materials for their application as ORR electrocatalysts. The synthesized catalyst exhibited better activity, stability, and resistance to methanol poisoning than the commercial Pt/C catalysts in alkaline media. EDS elemental mapping and XPS spectra confirmed the successful incorporation of F, N and Fe into the samples. The excellent ORR performance of our catalyst can be attributed to: (1) the highly electronegative F and N that changed the surface structure of carbon materials; (2) the morphology of 2_1_P_2_-Fe_1_-850 and its high content of active nitrogen (92.1%); (3) the co-doping of iron that further decreased the size of catalyst particles and increased the specific surface area of the catalyst (from 182 m^2^ g^−1^ of 2_1_P_2_ -850 to 1315 m^2^ g^−1^ of 2_1_P_2_-Fe_1_-850); and (4) the 3d orbital of iron. With the co-doping of iron, the charge of the active site can be adjusted to enhance the catalytic efficiency of the catalyst. This is consistent with our design idea. This study presents a general method for the mild preparation of F and other heteroatomic transition-metal-doped carbon materials and provides insights into the research and development of the ORR.

## Data Availability

The data generated or analyzed during the study are included in the article.

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
