# Peer review of "One-Step Synthesis of a Non-Precious-Metal Tris (Fe/N/F)-Doped Carbon Catalyst for Oxygen Reduction Reactions"

_molecules, 2023, doi:10.3390/molecules28052392_

Round 1

Reviewer 1 Report

The reviewed manuscript titled One Step Synthesis of a Non-Precious Metal Tris (Fe/N/F)-Doped Carbon Catalyst for Oxygen Reduction Reactions focused on the synthesis of a non-precious metal tris(Fe/N/F)-doped porous carbon electrocatalyst by a one-step process, using 2-methylimidazole, polytetrafluoroethylene, and FeCl3 as raw materials. Different ratios of precursors were studied to optimize the ratio of  2-methylimidazole and polytetrafluoroethylene, and the optimized condition were of the ratio 1:2 respectively (21P2-Fe1-850). The synthesized electrocatalyst had the enhanced electrocatalytic activity as compared to a commercial Pt/C in alkaline media. The paper is well written in scientific language and a well presented data with key-findings.

I recommend the following:

1.   In the abstract, line 21, after mentioning that the synthesized electro catalyst has a better methanol resistance than Pt/C, tell the reader what was this attributed to.(ie. Just before the sentence “This work…”).

2.      While Pt is the benchmark catalyst, many studies have been undertaken to replace the expensive Pt, as such, a motivation should come from there, as opposed to the blanket statement about the “expensive Pt” used.

3.      What does the “mildly doped” in the abstract mean?

4.      In Table 1, the ratios should be reported in % or molar ratios as in the text.

5.      It should be made clear what the source of carbon is.

6.      Describe the formation mechanism of the catalyst from pyrolysis.

7.      Line 187, Fig 3e is TEM image not STEM

8.      From the electron microscopy data, authors indicate that the sample without iron appears smooth and becomes fluffier with Fe-doping and mention that doping of transition metal iron can change the surface structure and morphology of heteroatom doped carbon materials. I agree with the authors, however, the expansion of this argument in the Raman data is not clear.

9.      In Figure 6, what is the peak between 2500 and 3000 Cm-1?

10. Give the surface area of 21P2-850 for comparison, as one wonder how the incorporation of Fe will affect it.

11.  What is the electrochemical surface area of 21P2-Fe1-850

12.  Line 232 and 233: Rephrase.

13.  In the conclusions, the authors give the factors affecting the electrocatalytic activity of the catalyst, however, it is still not clear what they attribute the enhanced performance to.

Reviewer 2 Report

In this work, the authors synthesized tris (Fe/N/F)-doped carbon materials as ORR electrocatalysts using a one-step sintering method. The ORR performance, stability, and methanol resistance of the catalysts were investigated and compared with the commercial Pt/C. However, the following comments need to be addressed before further consideration.

1.       The precursors were just hand-ground before the high-temperature treatment. I am concerned about the uniformity of the composition. The EDS elemental mapping is not enough to confirm the uniformity of the catalysts. Can the authors provide more evidence about uniformity?

2.       It is obvious that the iron source, nitrogen source, and fluorine source have different pyrolysis temperatures. For example, during the pyrolysis, 2-methylimidazole may melt first, then phase separation occurs or evaporate from the mixture, which may affect the final element ratio and the uniformity of the obtained catalysts. Thus, what’s the real element ratio of the catalysts? It shouldn’t the same as the initial element source ratio.

3.       Can the authors provide information about the size and size distribution of the particles?

Round 2

Reviewer 2 Report

The authors addressed the comments. It can be accepted now.